# A novel cyclic peptide (Naturido) modulates glia–neuron interactions *in vitro* and reverses ageing-related deficits in senescence-accelerated mice

Shinichi Ishiguro[1☯], Tetsuro Shinada[2☯], Zhou Wu[3,4☯], Mayumi Karimazawa[1], Michimasa Uchidate[5], Eiji Nishimura[2], Yoko Yasuno[2], Makiko Ebata[1], Piyamas Sillapakong[1], Hiromi Ishiguro[1], Nobuyoshi Ebata[1], Junjun Ni[3], Muzhou Jiang[3], Masanobu Goryo[6], Keishi Otsu[7], Hidemitsu Harada[7], Koichi Suzuki[1,6]*

1 Biococoon Laboratories, Inc., Ueda, Morioka, Japan, 2 Graduate School of Science, Osaka City University, Sumiyoshi-ku, Osaka, Japan, 3 Faculty of Dental Science, Department of Aging Science and Pharmacology, Kyushu University, Fukuoka, Japan, 4 Faculty of Dental Science, OBT Research Center, Kyushu University, Fukuoka, Japan, 5 Faculty of Science and Engineering, Iwate University, Ueda, Morioka, Japan, 6 Iwate University, Ueda, Morioka, Japan, 7 Division of Developmental Biology and Regenerative Medicine, Department of Anatomy, Iwate Medical University, Yahaba, Japan

☯ These authors contributed equally to this work.
* koichi@iwate-u.ac.jp

**Data Availability Statement:** All relevant data are within the manuscript and its Supporting Information files.

## Abstract

The use of agents that target both glia and neurons may represent a new strategy for the treatment of ageing disorders. Here, we confirmed the presence of the novel cyclic peptide Naturido that originates from a medicinal fungus (*Isaria japonica*) grown on domestic silkworm (*Bombyx mori*). We found that Naturido significantly enhanced astrocyte proliferation and activated the single copy gene encoding the neuropeptide *VGF* and the neuron-derived *NGF* gene. The addition of the peptide to the culture medium of primary hippocampal neurons increased dendrite length, dendrite number and axon length. Furthermore, the addition of the peptide to primary microglial cultures shifted CGA-activated microglia towards anti-inflammatory and neuroprotective phenotypes. These findings of *in vitro* glia–neuron interactions led us to evaluate the effects of oral administration of the peptide on brain function and hair ageing in senescence-accelerated mice (SAMP8). *In vivo* analyses revealed that spatial learning ability and hair quality were improved in Naturido-treated mice compared with untreated mice, to the same level observed in the normal ageing control (SAMR1). These data suggest that Naturido may be a promising glia–neuron modulator for the treatment of not only senescence, but also Alzheimer's disease and other neurodegenerative diseases.

## Introduction

Worldwide, fifty million people were living with dementia in 2018, and this number is expected to exceed 152 million by 2050 [1]. Moreover, in Japan, which has the highest

**Funding:** This study was supported by a grant from the Japan Society for the Promotion of Science to KS. DKS Co., Ltd. (the parent company of Biococoon Laboratories, Inc.) provided support for this study in the form of salaries for KS, SI, MK, ME, PS, HI and NE. The specific roles of these authors are articulated in the 'author contributions' section. The funders had no role in study design, data collection and analysis, decision to publish, or preparation of the manuscript.

**Competing interests:** The authors have read the journal's policy and the authors of this manuscript have the following competing interests: KS, SI, MK, ME, PS, HI and NE are paid employees of DKS Co., Ltd. (the parent company of Biococoon Laboratories, Inc.). This does not alter our adherence to PLOS ONE policies on sharing data and materials. There are no products in development or marketing products to declare. Biococoon Laboratories, Inc. and Lotte Co., LTD. have patents and those pending that are related to this study: JP/6182274, US/10654890, EP.FR.GB/ 3199541, DE/602015044038.2, TW/1683822, CN/ 201580051652.0 and KR/20177011000. Biococoon Laboratories, Inc. has patents pending that are related to this study: JP/2020-026373, CN/ 202010493307.5, KR/10-2020-0060224, JP/2020-038183 and JP/2020-183689

**Abbreviations:** AD, Alzheimer's disease; CGA, chromogranin; CNS, central nervous system; COF, coefficient of friction; IL-1β, interleukin-1β; Isaria japonica, *I. japonica*; Naturido, a novel cyclic peptide; SAMP8, senescence-accelerated model mice; SAMR1, normal ageing mice; TGF-β1, transforming growth factor-β1.

prevalence of dementia among Organisation for Economic Co-operation and Development (OECD) countries, prevalence of antidementia drugs (donepezil, galantamine, memantine, and rivastigmine) between April 2015 and March 2016 is 1.4% of the population, but 52% of clinical practices on dementia drugs have seriously focused on individuals with dementia who are younger than 85 years of age [2]. Recently, 132 agents have been or are being assessed in clinical trials for the treatment of Alzheimer's disease (AD) in the 2019 pipeline, but the development of AD drugs has a high failure rate. Thus, new treatments are urgently needed, and progress is being made in terms of defining new targets for the treatment of AD [3]. In addition, the cause of AD drug trial failure, with the unknown methodological limitations, and the appropriateness of therapeutic targets remain obvious problems in the search for new treatments [4].

Our previous study found by histochemical observation that an extract from a fungus (*Isaria japonica* = *Paecilomyces tenuipes*) grown on silkworm pupae (*Bombyx mori*) (Fig 1A) reversed astrogliosis in the hippocampal CA3 area and improved memory deficits in ageing mice [5,6]. These findings suggest that the extract of *I. japonica* fungus grown on silkworm was associated with recovery from central nervous system (CNS) deficits [7] and represents a promising astrocyte-targeted modulator. In addition, in a pilot study by our group, this fungal powder was administered as a nutraceutical to AD patients, which significantly increased the acetylcholine concentration in the cerebrospinal fluid that is a therapeutic target for symptomatic improvement in AD [8]. Thus, our previous evidence suggests that the extract of the fungus *I. japonica* may have potential as a new candidate drug for the treatment of senescence and neurodegenerative diseases.

Studying the interactions of both glia and neurons in the brain science of this century [9] would be essential to proceed with research on this extract in our experiments. Many studies on dementia and ageing have focused on glia in addition to neurons; in particular, astrocytes and microglia, which were long thought merely to support neurons, are now known to play distinct physiological roles in synaptic function, the blood–brain barrier and neurovascular coupling, as described in several reviews [10–14]. Astrocytes increase the numbers of mature, functional synapses between CNS neurons and are required for synaptic maintenance *in vitro* [15]. Additionally, microglia may act as multipotential stem cells that give rise to neurons, astrocytes and oligodendrocytes [16]. Zonisamide, which targets astrocytes to improve the cardinal symptoms of Parkinson's disease [17], and cannabinoids, which target cannabinoid receptors to abrogate microglia-mediated neurotoxicity during the pathology of AD [18], are now well known. However, the because glia–neuron interactions are extremely complex and instead of individual glia or neurons as in the past, new methodologies and strategic targets should be required for the development of effective preventive and therapeutic agents for AD and other neurological disorders represents a significant challenge. In other words, innovative approaches are needed to analyse these functional complexities and to identify therapeutic neuroprotective compounds.

Although previously we identified the extract of the fungus *I. japonica* may have potential as a breakthrough candidate, the active component of this extract was not identified in our previous works [5,6,8]. Screening agents that target glia and neurons described above, we aimed to highlight an active component from the extract and identified a promising candidate that might successfully modulate the viability and function of glia/neurons *in vitro*, that we refer to as Naturido. Additionally, the present study elucidated that the drug candidate enhanced astrocyte proliferation, anti-inflammatory effects, and neuron growth *in vitro*. Then, we investigated the effects of oral administration of the drug candidate on brain function and hair ageing in senescence-accelerated mice that we used in a previous study of ageing [19]. Taken together, our findings may provide a deep understanding of the effects of this new cyclic

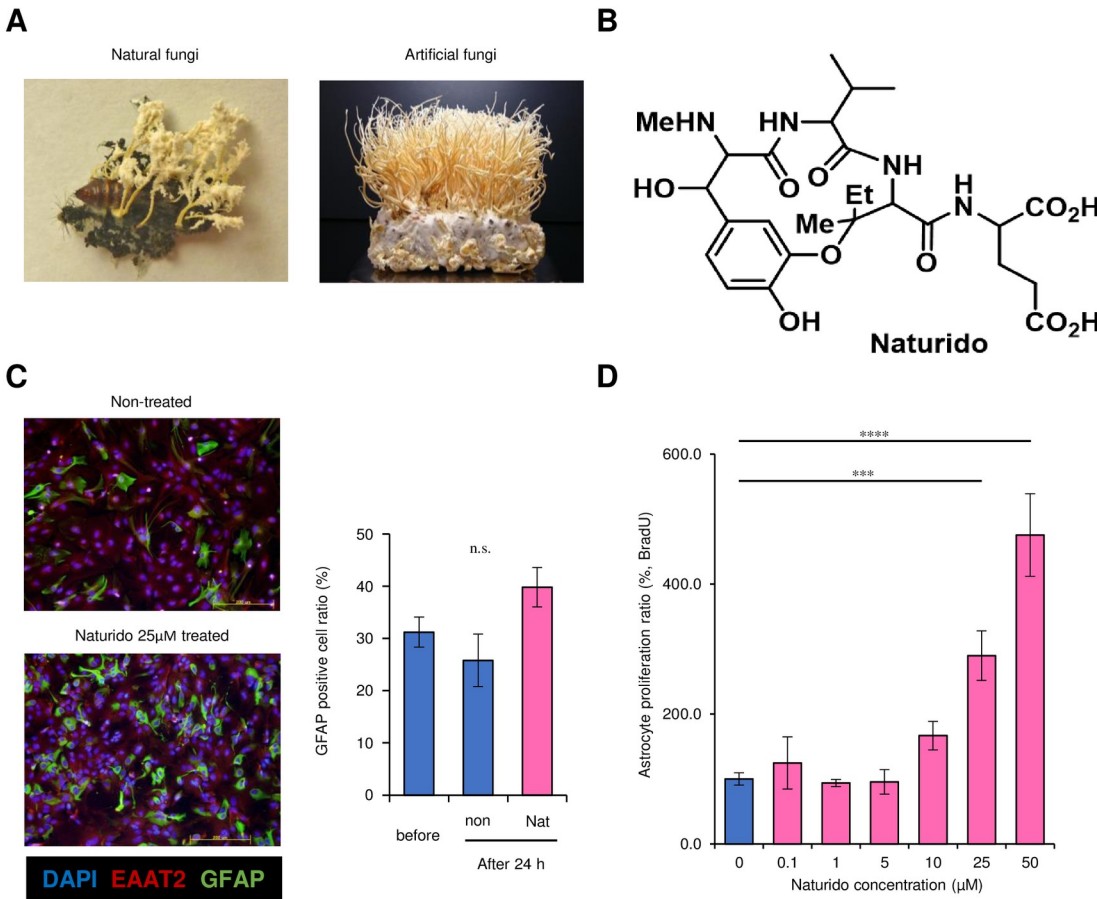

**Fig 1. *Isaria japonica*, the novel cyclic peptide (Naturido) extracted from it, and the effect of Naturido on astrocyte proliferation.** (A) Photographs of the fungus *Isaria japonica* collected in the field (left) and grown on silkworm pupae *(Bombyx mori)* in the laboratory at 25˚C for 65 days (right). (B) The chemical structure of Naturido purified from the *I. japonica* extract and analysed by NMR/MS (see Methods). (C) Histochemical observations (left panel) of primary cultured astrocytes not treated with Naturido (control, upper) and primary cultured astrocytes treated with 25 μM Naturido (lower) were carried out using DAPI for nuclear staining, anti-EAAT2 for astrocyte-specific staining, and anti-GFAP for reactive-type cell staining. The GFAP-positive cell ratio (right panel) of astrocytes (shown as a percentage of all astrocytes) was not significantly changed 24 h after Naturido treatment. (D) The effect of Naturido concentrations of 0.1 μM to 50 μM on the proliferative activity of primary cultured astrocytes, as measured with a BrdU assay. All values are expressed as means ± SEMs. ***$P < 0.001$, **** $P < 0.0001$ (C, vs before and D, vs 0 μM, Dunnett test using JMP 10.0.0).

peptide, which may be a useful tool for the development of new and effective therapies for CNS disorders.

## Materials and methods

### Animals

This study was carried out in accordance with the guidelines of the Animal Welfare Act and the Guide for the Care and Use of Laboratory Animals and was approved by the Animal Experiment Committee of Iwate University (A201127, A201428-1, A201413-2) and the Animal Care and Use Committee of Kyushu University (A29-134-1). Pregnant Jcl: ICR mice (originating from Institute of Cancer Research in U.S.A.) were purchased from SLC Japan (Shizuoka, Japan) and CLEA (Tokyo, Japan). Twenty-four- to 48-h-old neonatal and pregnant ICR mice as well as 8-week-old C57BL/6 male mice (SLC Japan) were used for *in vitro* bioassay

experiments. Nineteen-week-old male SAMR1 and SAMP8 mice were also purchased from SLC Japan and used for *in vivo* experiments. Three-day-old C57BL/6 mice were purchased from SLC Japan and kept in a specific pathogen-free environment at the Kyushu University Faculty of Dental Sciences according to protocols approved by the Animal Care and Use Committee of Kyushu University. A combination of cage number placement and markings with hair dye was used to discriminate among the mice. The applications of these mice were explained in S1 Fig. Total 75 animals were used but 15 animals that did not show any trial action were excluded in subsequent experiments.

## Isolation and structural determination

Fruiting bodies of *Isaria japonica* were artificially grown with dried and dead pupae of the silkworm *Bombyx mori* for its medicinal properties at 25°C for 65 days, and they were then lyophilized and crushed. The extracts of *I. japonica* have potential neuroprotective effects, as some have been found to prevent astrogliosis in the CA3 area of the hippocampus in aged mice [5,6]. Thus, our first experiment focused on the isolation and structural determination of *I. japonica*, a fungus with the ability to promote astrocyte proliferation that has physiological activity.

**Hot water extraction.** *I. japonica* powder, which was provided by Tohaku Nosan Kigyo Kumiai (currently known as Biococoon Laboratories, Inc.) in the Fukushima Prefecture in Japan, was prepared as described above and used in the following experiments. Ten times (w/v) the amount of Elix water (EW) (420 ml) was added to the powder (42 g), and a hot water extract was obtained after heating at 120°C for 20 min in an autoclave. After that, the hot water extract was filtered with qualitative filter paper (No. 2 ADVANTEC) and collected. Ten times the amount of EW (420 ml) was again added to the residue after the first filtration, and a second extraction and filtration procedure was performed under the same conditions. These liquids were combined, filtered again, and subjected to freeze drying using a freeze dryer (EYELA FDU-2100, manufactured by Tokyo Rikakikai). The obtained powder was used throughout the remainder of the experiments as the *I. japonica* hot water extract (hereafter referred to as the IJE) and stored at −80°C until use.

**Two-phase distribution.** To isolate astrocyte-targeting modulators on the base of our previous finding that the IJE might have reversable potentials of astrogliosis and memory deficits in ageing mice [5,6], first, six grams of the IJE obtained from the step described above was dissolved in 50 times the amount (w/v) (300 ml) of EW, and the mixture was then transferred to a separatory funnel. By shaking the separatory funnel, the IJE was evenly distributed within the funnel, and then 300 ml of ethyl acetate was suspended. After repeated shaking and gas discharge 10 min thereafter, the separatory funnel was allowed to stand for 60 min for separation of the mixture into 2 layers. The bottom EW layer (EW-1) was collected; then, 300 ml of EW was added to the ethyl acetate layer that remained in the separatory funnel, and the same procedures as those described above were repeated. The aqueous layer (EW-2; the secondary bottom layer) and the ethyl acetate layer (EA-1; the upper layer) were collected. After the addition of EW-1 (the first layer collected) and 300 ml of ethyl acetate to the separatory funnel, the same procedures as those described above were repeated. EW-1, which formed a new bottom layer, and the secondary ethyl acetate layer (EA-2) were collected. In a two-phase distribution, the mixture of EW-1 and EW-2 was used as the EW fraction, and the mixture of EA-1 and EA-2 was used as the ethyl acetate fraction. Each fraction was subjected to condensation drying by using a series of rotary evaporators (CCA-1100, DPE-1220, SB-1000, N-1000, EYELA DTLT-20, ULVAC) and a freeze dryer (EYELA FDU-2100). The obtained powders were collected as either the EW layer extract or the ethyl acetate extract of the two-phase distribution and were stored in an ultra-low-temperature bath at −80°C until use.

**Reverse-phase (RP) flash column chromatography.** To further purify the EW layer extract of the two-phase distribution obtained in the step described above, RP flash column chromatography was carried out. The column was prepared by a dry-type filling method. Briefly, silica gel (Wakosil 40C18, Wako Pure Chemical Industries, Ltd.) was added as a carrier to a flash chromatography column to yield a column volume of 160 cm$^3$. After methanol-induced swelling for making firm carrier and the addition of EW as an initial development solution to the top portion of a 500 ml chromatography column, the solvent and air inside the carrier were extruded with pressure using a pump (HIBLOW air pump, type SPP-6EBS, Techno Takatsuki Co., Ltd.).

A total of 3.5 mg of the EW layer extract described above was dissolved in 14 ml of EW and charged onto the carrier surface of the prepared column. Pressure was applied by using a pump, and 500 ml of EW and 300 ml each of 10%, 20%, 40%, 60%, 80%, and 100% MeOH were sequentially allowed to flow through the column. Each of the eluted fractions for physiological assay was subjected to concentration drying by using a series of rotary evaporators and a freeze dryer, as described above. The obtained fractions were named F1 (the MQ extract fraction), F2 (the 10% MeOH fraction), F3 (the 20% MeOH fraction), F4 (the 40% MeOH fraction), F5 (the 60% MeOH fraction), F6 (the 80% MeOH fraction), and F7 (the 100% MeOH fraction) and then stored at −80˚C until use.

The physiological activity of each of the obtained fractions (F1 to F7) was determined by preparing astrocytes and assessing astrocyte proliferation used the measurements of BrdU incorporation as mentioned below, and it was confirmed that the components with the ability to promote astrocyte proliferation were contained in the F3 fraction.

**Purification by RP–and hydrophilic interaction liquid chromatography (HILIC)–high-performance liquid chromatography (HPLC).** The F3 fraction obtained in the step described above was purified with four subsequent steps: steps 1 to 3 involved RP-HPLC with a Develosil column, and step 4 involved HILIC. For step 1, a Develosil RPAQUEOUS column (20.0 inner diameter [ID] × 250 mm) (Nomura Chemical Co., Ltd.) was used. The column temperature was 40˚C, and the mobile phase consisted of an EW/MeOH solution that varied with time. The time course was as follows (where % indicates the percentage of MeOH in the mobile phase): 0 to 60 min, isocratic at 1.0%, 5.0 ml/min; 60 to 180 min, gradient of 1% to 30.4%, 2.0 ml/min; 180 to 212 min, gradient of 30.4% to 100%, 5.0 ml/min; and 212 to 292 min, isocratic at 100%, 5.0 ml/min. The detection wavelength was 254 nm. For steps 2 and 3, the same column (Develosil RPAQUEOUS, 20.0 ID × 250 mm) and temperature (40˚C) were used. The mobile phase consisted of EW with 0.01% acetic acid and MeOH with 0.01% acetic acid, and the flow rate was 5.0 ml/min. The time course was as follows (where % indicates the percentage of MeOH with 0.01% acetic acid in the mobile phase): 0 to 30 min, isocratic at 1%; 30 to 70 min, gradient of 1.0% to 40.0%; and 70 to 100 min, isocratic at 100%. The detection wavelength was 254 nm. For step 4, a HILIC column (4.6 × 250 mm) (COSMOSIL) was used with a column temperature of 28˚C, and the mobile phase was composed of solvents A (20 mM Et$_2$ NH-CO$_2$ buffer, pH 7.0) and B (CH$_3$CN) (A:B = 90:10). Isocratic elution was performed with a flow rate of 1.0 ml/min, and the detection wavelength was 210 nm.

After the 4 purification steps were carried out and after assessing astrocyte proliferation, the F3-10, F3-10-4, F3-10-4-5, and F3-10-4-5-3 fractions were determined to contain the active components and were collected. Each of the obtained fractions was subjected to concentration drying by using one type of rotary evaporator, as described above; then, the final purified products were obtained as bis (diethylamine) salts. It was also confirmed that the components with the ability to promote astrocyte proliferation were contained in the final purified product of the F3-10-4-5-3 fraction, as shown in S1 Table and Fig 1A.

## Method for structure determination of Naturido

One and two-dimensional nuclear magnetic resonance spectroscopy were recorded on Bruker AVANCE 600 (600 MHz) spectrometer. Chemical shifts of $^1$H NMR were reported in parts per million (ppm, d) relative to HDO (d = 4.70) in $D_2O$. Low resolution mass and high-resolution mass spectrum were obtained on a JEOL JMS-700T for fast atom bombardment ionization (FAB) [positive mode, matrix: glycerol]. Optical rotations were taken on a JASCO P-1030 polarimeter with a sodium lamp (D line). Amino acid analysis was performed by an enantiomer labeling method of amino acid analysis (110˚C, 24 h) at HiPep Laboratories. The analytical data are informed in Supplemental Results.

## Cell culture experiments

**Primary culture of cerebral astrocytes and proliferation assay.** Neonatal ICR mice (24 h to 48 h old) were sufficiently sterilized with 70% ethanol and immersed in 30 ml of PBS in a 100 mm cell culture dish (True Line), and the dish was then placed on a clean bench. The neonatal mice were euthanized by cervical dislocation using tweezers. The skull of each mouse was opened, and the entire brain was removed. Each obtained brain was transferred to a 100 mm cell culture dish containing 15 ml of high-glucose Dulbecco's modified Eagle medium (HG-D-MEM, FUJIFILM Wako Pure Chemical Corporation). By using tweezers, the olfactory bulb, median eminence, and meninges were removed, leaving only the cerebrum. Subsequently, the cerebrum was transferred to a 100 mm cell culture dish containing 10 ml of HG-D-MEM; using a scalpel, the cerebrum was finely cut into < 1 $mm^2$ pieces. After cutting, the cerebrum was transferred together with the medium to a 50 ml conical tube (Techno Plastic Products AG). After the tube was allowed to stand for 2 min, the supernatant was removed. Then, 4 ml of fresh HG-D-MEM, 400 μl of 2.5% trypsin (Sigma-Aldrich) and 40 μl of 1% DNase (Sigma-Aldrich) were added to the tissue remaining in the tube, and the mixture was incubated in a water bath at 37˚C for 10 min with intermittent stirring. Subsequently, 10 ml of HG-D-MEM (10% FBS) was added to the cerebral tissue to terminate the trypsin reaction, and the mixture was centrifuged for 3 min at $1,000 \times g$ (Lex-100, Tomy Seiko Co., Ltd.). The supernatant was collected with an electronic pipette. Then, 10 ml of HG-D-MEM (10% FBS) was added to the pelleted cell mass, and the cell mass was resuspended by pipetting up and down with a sterile pipette several times. To remove the remaining cell mass, the suspension was passed through a cell strainer (pore diameter of 70 μm, Falcon), and the cells that passed through the strainer were counted by using a cell counting plate. Before cell seeding, a 100 mm dish was coated with laminin (Sigma-Aldrich, 1 μg/$cm^2$ culture area) and fibronectin (Sigma-Aldrich, 3 μg/$cm^2$ culture area) to prevent cell-peelings, washed 3 times with PBS and maintained at 5˚C. After adjusting the density of cells ($8.7 \times 10^4$ cells/$cm^2$), the suspended cells were seeded onto the laminin- and fibronectin-coated 100 mm dishes (8 ml per plate) and incubated at 37˚C under 5% $CO_2$. Ninety-six hours later, the medium was aspirated once. After briefly washing the inside of each 100 mm dish with 10 ml of PBS, 8 ml of fresh HG-D-MEM (10% FBS) was added. The method described above, and the astrocyte preparation method described below were based on previously described methods [20].

Seventy-two hours after the medium exchange described above, the 100 mm dishes with cell medium were removed from the incubator, tightly covered with Parafilm and prepared in sets of 3 to 4 dishes. The cells were cultured with agitation on a rotary shaker (WB-101 SRC, WakenBtech Co., Ltd.) at 100 rpm at a temperature of 37˚C for 20 h, and the neuronal cells, cell debris, dead cells and other components were removed from the dish. Only astrocytes, which are a type of glial cell in the central nervous system (CNS), adhered to the 100 mm dishes (hereafter, these cells are called primary cultured astrocytes). After shaking, the 100 mm

dishes were transferred to a clean bench, the supernatant was removed by aspiration, and the dishes were washed with 10 ml of PBS. After adding 2 ml of 0.25% trypsin (Sigma-Aldrich) using a Pasteur pipette, each dish was left to stand in an incubator for 10 min. The 100 mm dishes were again brought back to the clean bench, and 10 ml of low-glucose Dulbecco's modified Eagle medium (LG-D-MEM, FUJIFILM Wako Pure Chemical Corporation) (10% FBS) was added. After terminating the trypsin reaction, the cell medium was collected in a 50 ml conical tube. After that, the number of cells was counted using a cell counting plate, and the cell density was adjusted to $2.2 \times 10^4$ cells/$cm^2$ with LG-D-MEM (10% FBS). The resulting astrocytes in the cell broth were seeded onto laminin- and fibronectin-coated 100 mm dishes (8 ml of broth per dish).

The cultured astrocytes prepared as described above were cultured for 2 weeks with medium exchange every 72 to 96 h after seeding. The procedures for the medium exchange were essentially the same as those used in the above experiment except that LG-D-MEM (10% FBS) was used as the medium. Fourteen days later, the number of cultured astrocytes was adjusted according to the method described above, and subculture was carried out.

A suspension of the primary cultured astrocytes was prepared with a density of $6.3 \times 10^4$ cells/$cm^2$ in the same order as that described for the isolation and purification of the components with the ability to promote astrocyte proliferation. LG-D-MEM (10% FBS) was used as the medium. The cultured astrocytes in suspension were seeded at a volume of 100 μl/well into 96-well microplates (treated polystyrene tissue culture plates, AGC Techno Glass Co., Ltd.) for cell culture by using a multi-pipette (Eppendorf) and were incubated under conditions of 37˚C and 5.0% $CO_2$. Twenty-four hours later, to suppress astrocyte proliferation, the medium in each well was replaced with LG-D-MEM (0% FBS); 24 h after that, the astrocytes were exposed to Naturido concentrations of 0.1 μM to 50 μM and other samples dissolved in LG-D-MEM (0% FBS) for 24 h. LG-D-MEM (0% FBS) without sample was used as a control. Major subsequent experiments were assayed at a 25 μM Naturido concentration. After exposure to the sample, a BrdU colorimetric ELISA (Roche Diagnostic GmbH) was performed to assess cell proliferation based on the kit protocol. Absorbance at 370 nm was measured on a microplate reader to assess proliferation ability (Infinite M200 Pro, Tecan Group, Ltd.). In this test, the reaction time for the BrdU labelling solution was set at 4 h, the reaction time for the peroxidase (POD)-labelled anti-BrdU antibody reaction solution was set at 2 h, and the reaction time for the substrate solution was set at 30 min without the use of a stop solution. Furthermore, tapping of the plate was avoided throughout the test to prevent cell loss at each step, and removal of the medium or chemical reagent was carried out by using a multi-pipette.

**Identification of primary cultured astrocytes and determination of the characteristics of the obtained cells.** To exclude the possibility of contamination of the primary cultured astrocytes with microglia or oligodendrocytes derived from either neuronal cells or glial cells, an immunohistochemical analysis was carried out with antibodies specific to each of these cell types. First, secondary subcultured astrocyte suspensions were prepared in a manner similar to that described above at a density of $6.7 \times 10^3$ cells/$cm^2$. LG-D-MEM (10% FBS) was used as the medium. The prepared cultured astrocyte suspensions were added to 35 mm dishes (BD Falcon) whose surfaces were coated with laminin (Sigma-Aldrich, 1 μg/$cm^2$ culture area) and fibronectin (Sigma-Aldrich, 3 μg/$cm^2$ culture area) at a volume of 2 ml/dish. The cells were then cultured under conditions of 37˚C and 5.0% $CO_2$. After 24 h of culture, the medium in each dish was replaced with LG-D-MEM (0% FBS) to suppress astrocyte proliferation. After further culture for 24 h under conditions of 37˚C and 5.0% $CO_2$, the medium in each dish was replaced with LG-D-MEM (0% FBS) containing Naturido dissolved in advance to a concentration of 25 μM. Then, the astrocytes were exposed to Naturido for 0 or 24 h under conditions of 37˚C and 5.0% $CO_2$. LG-D-MEM (0% FBS) without the addition of Naturido was used as a control.

All of the medium was removed from the dishes in each treatment group by aspiration. After adding a PBS solution containing 4% paraformaldehyde (Wako) and shaking for 15 min at room temperature, the cells were fixed. Then, a PBS solution containing 0.1% Triton X-100 (Sigma-Aldrich) was added, and the cells were permeabilized by shaking for 5 min. After permeabilization, Image-i T FX Signal Enhancer (Life Technologies) was added, and the blocking by this enhancer was carried out with shaking for 1 h at room temperature.

Subsequently, for the identification of nuclei and cell types, DAPI for nuclear staining and primary antibodies (anti-MAP2 [a neuronal marker] produced in chickens, anti-EAAT2 [an astrocyte marker] produced in sheep, anti-GFAP [anther astrocyte marker] produced in chicken, anti- CD11b/c [a microglial marker] produced in mice, and anti-MBP [an oligodendrocyte marker] produced in mice; all manufactured by Abcam) were added to the cells after blocking, followed by shaking for 1 h at room temperature. To detect the primary antibodies, secondary antibodies (Invitrogen) conjugated to the fluorophore Alexa Fluor 488 or Alexa Fluor 546 were added, and the cells were shaken at room temperature for 30 min. Fluorescence was observed, and images were photographed by using an IX71 fluorescence microscope (Olympus). The results are shown in Fig 1C and S3 Fig.

**Comparison of astrocyte proliferation and acetylcholinesterase (AChE) inhibition between Naturido and other therapeutic agents.** To investigate whether therapeutic drugs (zonisamide, FUJIFILM Wako Pure Chemical Corporation; donepezil HCl, Sanyo Chemical Laboratories Co., Ltd; eserine, Sigma-Aldrich; galantamine, Abcam) have astrocyte proliferative activity, astrocytes were prepared with a density of $6.3 \times 10^4$ cells/cm$^2$, and zonisamide as another control and three AChE inhibitors were used at the same concentration as Naturido (25 μM). Astrocyte proliferative activity was measured with a BrdU assay (%). On the other hand, AChE that plays a role in the hydrolysis of the neurotrasmitter has proven to be an important therapeutic target for AD and AChE inhibitors have been developed for clinical treatments [21]. Here, we also tested the relationship between Naturido and AChE inhibition. The inhibitory activity on acetylcholinesterase was measured according to a standard method [22], and the concentration of donepezil HCl, used for comparative purposes, was determined with a previously described method [21].

**Primary culture of hippocampal neurons.** A basic protocol was modified [23] by the reason of appropriate cell numbers in our experiments. Pregnant mice were killed the day before giving birth using an approved method of euthanasia, and their uteruses were removed and placed in TPP dishes (100 mm) sterilized with 95% ethanol. On a clean bench, the uteruses were placed in dishes with PBS, and the brains were removed from the foetuses and placed in TPP dishes (100 mm) containing CMF-HBSS as described in detail previously [23]. The brains remained submerged in medium at all times. Under a dissecting microscope, the cerebral hemispheres were removed, and the meninges were carefully stripped away with fine forceps. The same hippocampi were collected in 35 mm dishes containing 2 ml of CMF-HBSS. All of these tissues were then removed and placed in 15 ml conical tubes, and the total volume was brought to 4.5 ml with CMF-HBSS.

Then, 0.5 ml of 2.5% trypsin was added to the tissues, which were incubated in a water bath at 37˚C for 15 min. After gently removing the supernatant and the addition of 5 ml of CMF-HBSS, the hippocampi were left at the bottoms of the tubes, which were kept at room temperature for 5 min. These tissues were also washed twice with 5 ml of CMF-HBSS per wash. After the second wash, the final volume was brought to 3 ml with CMF-HBSS. The hippocampi were dissolved by repeatedly pipetting up and down with a Pasteur pipette: first, 5 to 10 passes were conducted with a regular pipette, and then 5 to 10 passes were conducted with a Pasteur pipette that had been flame-polished so that its tip diameter was narrowed by half. The suspension was expelled forcefully against the wall of the tube to minimize foaming and

tissue chunks. After filtration with a 70 μm mesh (Falcon) and addition of a drop of the cell suspension to a haematocytometer (Nanoentek, C-Chip), $1.0$–$2.0 \times 10^5$ cells/hippocampus was determined as the total yield. Trypan blue staining revealed that the cell viability was above 90%. Neuronal Plating Medium (NPM) was added to the neuronal cell culture. Using a micropipette, $3.5 \times 10^5$ cells per 60-mm dish with a bottom area 21 cm$^2$ (viz., $1.4 \times 10^4$ cells/cm$^2$) (True Line) were added to each dish containing 5 polylysine-treated coverslips (Fisher Scientific, Microscope Cover Glass) [23]. After 3 h on the cell seeding, each coverslip was gently inverted with fine forceps and transferred to a 12-well plate, the wells of which already contained 1 ml each of Neurobasal/B27 and the test agent (Naturido). After 3 days of culture, phase-contrast images of the neurons were obtained with BZ-X710 and BZ-Viewer (Keyence). Using a BZ-X analyser, the dendrite total length and number and axon length were evaluated according to a method described previously [24].

**Proliferation test for cells other than astrocytes and neurons.** To test whether Naturido-mediated promotion of proliferation is specific to astrocytes or is common to other cells, we chose popular tumour and normal cells. The influence of Naturido on the proliferation of C6 glial tumour cells (cell line nos. JCRB9096, NIBIOHN), normal human dermal fibroblasts (NHDFs, Kurabo Industries, Ltd.), human hepatocellular carcinoma (HepG2, Cosmo Bio Co., Ltd.) cells, and human myeloid leukaemia (K562, Cosmo Bio Co., Ltd.) cells was measured by WST-1 assay with a previously described method [25]. Each cell type was seeded on a 96-well plate (treated polystyrene tissue culture plates for C6 glial tumour cells, NHDFs and HepG2; non-treated polystyrene tissue culture plates for K562, AGC Tech Glass Co., Ltd.) in 100 μl of cell suspension. NHDFs and C6 glial tumour cells were prepared at a density of $7.8 \times 10^3$ cells/cm$^2$, while HepG2 and K562 cells were prepared at a density of $15.6 \times 10^3$ cells/cm$^2$. Because Naturido was added after being dissolved in PBS, PBS solution only was used as a control.

## Analysis of astrocyte gene expression on addition of Naturido

We chose representative genes of neurotrophic factors related to astrocyte–neuron interactions. The influence of 25 μM Naturido on the expression of the genes nerve growth factor (*NGF*), glial cell line-derived neurotrophic factor (*GDNF*), vascular endothelial growth factor (*VEGF-A*), brain-derived neurotrophic factor (*BDNF*), and VGF nerve growth factor inducible (*VGF*), which encode representative neurotrophic factors, was determined over time in primary cultured astrocytes.

Astrocytes were cultured with the same method as that described above, except that the suspension of primary astrocytes was prepared with a density of $6.7 \times 10^4$ cells/cm$^2$ astrocytes were exposed to 25 μM Naturido for 0, 1, 2, 4, 8, 12, and 24 h. LG-D-MEM (0% FBS) without the addition of Naturido was used as a control for each exposure duration. Then, using an RNeasy Mini Kit (Qiagen), the extraction and purification of total RNA from the cell suspensions were carried out. The total extracted RNA was quantified by using a NanoPhotometer (Implen). cDNA was synthesized through reverse transcription by reacting 500 ng of total RNA with PrimeScript RT Master Mix (TaKaRa) at 37˚C for 15 min using a GeneAmp PCR System 9600 (PerkinElmer), and the reaction was terminated with heating at 85˚C for 5 s.

Analysis of gene expression was carried out by using the synthesized cDNA as a template, commercially available primers (obtained from TaKaRa; see S2 Table), SYBR premix Ex Taq I (TaKaRa) and a real-time PCR device, the Thermal Cycler Dice TP800 (TaKaRa). The primers comprised the following: a primer for *Mus musculus Ngf* transcript variant 1 mRNA (MA07578), a primer for *Mus musculus Gdnf* mRNA (MA102345), a primer for *Mus musculus Vegfa* transcript variant 1 mRNA (MA128545), a primer for *Mus musculus Bdnf* transcript variant 2 mRNA (MA138332), and a primer for *Mus musculus Vgf* mRNA (MA157656). The

expression level of each target gene was compared after calibration against an internal standard, the housekeeping gene *GAPDH*.

## Microglial cell culture, determination of cell viability, and real-time quantitative PCR (RT-PCR) analysis

**Primary microglial culture.** CD11b$^+$ cells were isolated from the brains of adult mice (8 weeks old, male) with the magnetic-activated cell sorting (MACS) method. The brains were then cut into small pieces. The tissues were further mechanically dissociated by a gentle MACSDissociator (Miltenyi Biotec) with enzymatic digestion using a Neural Tissue Dissociation Kit (\Biotec). After filtering through a 30 mm cell strainer, single-cell suspensions were obtained. After magnetic labelling with CD11b MicroBeads, the cell suspensions were loaded onto magnetic MACS columns placed in a magnetic separator (Miltenyi Biotec). After rinsing the MACS columns with PBS, the CD11b$^+$ cell fraction was collected according to previously described methods [26].

**Determination of microglia viability.** Primary microglia ($3.1 \times 10^4$ cells/cm$^2$) were cultured in a 96-well plate for 24, 48, and 72 hand then incubated with concentrations of Naturido (0.01, 0.1, 1 and 5mM) for 72 h. Cell viability was assessed using a Cell Counting Kit-8 (CCK-8; Dojindo Molecular Technologies, Inc., Kumamoto, Japan) in accordance with the manufacturer's instructions, as follows: after treatment with Naturido, 10 μl of CCK-8, was added to each well of the 96-well plate, and the plate was then incubated at 37˚C for 2 h. In accordance with the instructions, the optical density was read at a wavelength of 450 nm using a microplate reader. Cell viability was calculated using the following formula: optical density of the treated group / optical density of the control group × 100%).

**Microglia RT-PCR analysis.** mRNA was isolated from primary microglia ($3.1 \times 10^4$ cells/cm$^2$) after stimulation with chromogranin-A (CGA) (synthetic human CGA286-301, Peptide Institute, Osaka, Japan), or after pre-treatment with Naturido according to previous described methods [26]. Total RNA was extracted with RNAiso Plus (TaKaRa, Japan) according to the manufacturer's instructions. A total of 800 ng of extracted RNA was reverse transcribed into cDNA using a QuantiTect Reverse Transcription Kit (Qiagen, Japan). After an initial denaturation step at 95˚C for 5 min, temperature cycling was initiated. Each cycle consisted of denaturation at 95˚C for 5 s, annealing at 60˚C for 10 s, and elongation for 30 s. In total, 40 cycles were performed. The cDNA was amplified in duplicate using a Rotor-Gene SYBR Green RT-PCR Kit (Qiagen, Japan) with a Corbett Rotor-Gene RG-3000A Real-Time PCR System. The data were evaluated using the RG-3000A software program (version Rotor-Gene 6.1.93, Corbett). The sequences of the primer pairs are as follows: *IL-1β*: 5'-CAACCAACAAGTGAT ATTCTCCATG-3' and 5'-GATCCA CACTCTCCAGCTGCA−3'; *TGF-β*, 5′-TCAGACATTCGG GAAGCAGTG-3′ and 5′-ATTCCGTCTCCTTGGTTCAGC-3′. For data normalization, an endogenous control (*actin*) was assessed to control for the cDNA input, and the relative units were calculated with the comparative Ct method.

## Assessment of learning and memory in a senescence-accelerated mouse model after oral administration of Naturido

In order to examine the effects of Naturido *in vivo*, a mouse model utilizing senescence-accelerated mouse (SAMP8), which has been used as a model for cognitive disorders associated with ageing, was selected. SAMR1 mice under normal ageing conditions were selected as normal controls [19,27,28], which have the characteristic features of the retention in learning and memory and hair anti-ageing. In our previous study [5], it was confirmed that the aged mice orally administrated either concentration of 2.5 or 25 mg/kg/ day of the *I. japonica* extract

improved the spatial learning ability. We here designed lower concentrations of 2.5 and 25 μg/kg/day in the oral administration of Naturido identified from the extract. Nineteen-week-old male SAMP8 and SAMR1 mice were randomly divided into 5 groups on the base of standard mouse weights: the untreated SAMR1 group (36.78 ± 0.73 g), the untreated SAMP8 group (30.67 ± 1.02 g), the SAMP8 + 1250 μg donepezil HCl/kg/day (positive control) group (30.10 ± 0.89 g), the SAMP8 + 2.5 μg Naturido/kg/day test group (27.40 ± 0.56 g), and the SAMP8 + 25 μg Naturido/kg/day test group (27.88 ± 0.75 g). The animals were maintained as follows: one animal/cage under constant conditions, with a room temperature of 24 ± 1˚C, with the lights on from 7:00 to 19:00 (darkness from 19:00 to 7:00), and with a humidity of 50 ± 10%. The experiment was carried out from 13:00 to 18:00 in an incubator under constant conditions. After an acclimation period of 10 days, all mice were provided with a standard diet (MEQ, Oriental Yeast Company), and the body weights of the animals were recorded every day during the entire test period. The amount of food and water consumed as well as the excretion amounts were weighed 2 times per week. Each individual mouse was identified based on cage number.

In terms of treatment, the SAMR1 group (n = 10) was administered 0.9% physiological saline via an animal feeding needle for 5 weeks. The SAMP8 group (n = 15) was administered 0.9% physiological saline for 5 weeks. However, oral administration of saline was continued until the whole behavioural test was over; therefore, administration was continued for 8 weeks in total. The SAMP8 + 1250 μg donepezil HCl/kg/day group (n = 14 in the step-through passive avoidance learning test; n = 11 in the Morris water maze test) was administered a concentration of 1250 μg donepezil/kg/day via a gastric tube for 5 weeks. Oral administration was also continued in this group until the whole behavioural test was complete, for a total administration time of 8 weeks. The SAMP8 + 2.5 μg Naturido/kg/day (n = 16 in the step-through passive avoidance learning test; n = 14 in the Morris water maze test) and the SAMP8 + 25 μg Naturido/kg/day groups (n = 15 in the step-through passive avoidance learning test; n = 7 in the Morris water maze test) were administered the two concentrations of Naturido via an animal feeding needle for 5 weeks.

**Step-through passive avoidance learning test.**   This test was carried out to investigate the contextual learning ability of mice [5,6]. The step-through passive avoidance learning test utilizes the negative phototaxis of a mouse, i.e., the preference of a mouse for a dark place over a light place. The apparatus (O'Hara & Co., Ltd.) consisted of a light box and a dark box, each of which had an inverse trapezoidal shape (light box: top surface 100 × 130 mm, bottom surface 42 × 130 mm, height 90 mm; dark box: top surface 100 × 160 mm, bottom surface 42 × 160 mm, height 90 mm). On the floors of both boxes, stainless-steel bars with diameters of 2 mm were arranged at intervals of 6 mm; however, the bottom surface of only the dark box could be electrically charged. The two boxes were separated by a partition plate that could be freely opened and closed in the vertical direction by the person who performed the test. To ensure that abnormal mice were excluded from the main test, a pre-acquisition trial was carried out. For the pre-acquisition trial, a mouse was placed in the light box with white fluorescent light (400 lux), while the partition plate was left open, and the time until the mouse entered the dark box was measured. In each acquisition trial for the memory retention, a mouse was placed in the light box, which was illuminated, and the test was initiated by opening the partition plate. Infrared (IR) sensors were located on the left and right sides of the entrance of the dark box to detect mouse entry; the signals detected by these IR sensors were used for the measurement of entry time or as triggers to generate an electric shock.

Given that mice prefer darkness over light, if a mouse placed in the light box stayed in that box for 60 s or longer, the animal was considered abnormal and was thus not used in subsequent experiments. On the 1st day of the test, a pre-acquisition trial was performed, and an acquisition trial was performed 15 min later. For the acquisition trial, a mouse was placed in the

light box while the partition plate was closed. Thirty seconds later, the partition plate was opened, and the time until the mouse entered the dark box (i.e., the latency) was measured. The partition plate was closed when the rear paw of the mouse entered the dark box or when an IR sensor in the dark box was activated, and 2 s after the mouse entered the dark box, an electric shock of 0.3 mA was applied for 4 s. On the 2nd day of the test (i.e., 24 h after the acquisition trial), a post-shock trial was performed. The post-shock trial was performed with the same procedures as those in the acquisition trial except that, unlike in the acquisition trial, an electric shock was not applied. The latency in the post-shock trial was measured, with a maximum of 300 s.

**Morris water maze test.** The Morris water maze test for the investigation of spatial learning ability was performed one week after the step-through passive avoidance test that is described above. A circular pool (with a diameter of 100 cm and a depth of 30 cm) was set 80 cm above the bottom of the apparatus (O'Hara & Co., Ltd.). Subsequently, water was added to the pool to a depth of 20 cm (water temperature $25 \pm 1°C$), and a transparent platform (with a diameter of 10 cm and a height of 19 cm) was set such that it was immersed 1 cm below the water surface. Next, the water in the pool was clouded with commercially available white poster paint so that the platform remained invisible to the mice while swimming. A photographic image covering every quadrant was automatically taken by a black-and-white CCD camera 100 cm above the water surface at the centre of the pool. The camera was connected to a computer, and the swimming path of each mouse was determined at intervals of 0.5 s. Image WMH 2.08 and Image WM 2.12 (O'Hara & Co., Ltd.), which are software programs based on the Image program developed and published by the U.S. National Institutes of Health (NIH), were used to record the swimming paths and analyse the images.

The procedure used for the Morris water maze test is also based on previous methods [5,6]. The test was initiated at the same time every day for 9 days. On the first day, each mouse was allowed to swim once for 1 min so that it could familiarize itself with the pool. Thereafter, a label with a height of 10 cm was set on the platform, to enable the mouse to recognize the presence of the platform. At the beginning of the trial, each mouse was placed in the water at a random location selected by the computer facing the wall of the pool, and the person conducting the test rapidly moved away from view of the mouse. If the mouse reached the platform within 60 s, the mouse was allowed to stay on the platform for 15 s and was then retrieved. If the mouse could not reach the platform by swimming within 60 s, the mouse was transferred to the top of the platform by hand by the person running the test and then retrieved after being allowed to stay on the platform for 15 s.

From the second to the eighth day, training was conducted to allow each mouse to memorize the location of the platform. Training was continued 4 times per day for each mouse. The training method followed the same procedure as that used on the first day, and the time to reach the platform was recorded. If the mouse could not reach the platform after swimming for 60 s, the mouse was transferred to the top of the platform by hand by the person running the test and allowed to stay there for 15 s, and the time to reach the platform was recorded after 60 s. For the probe test, the platform was removed from the pool. After allowing each mouse to swim for 60 s, the visit rate for each domain of each quadrant (1/4 of the circular pool) was measured. The probe test was carried out once for each mouse on the ninth day. Mice that did not show any trial action were not used in this test.

## Hair anti-ageing effects observed in the senescence-accelerated mouse model after oral administration of Naturido

According to our previous method [19], we analysed the body hair-related anti-ageing effects of Naturido in the senescence-accelerated mouse model by means of topographical analysis

and friction measurement, which are known to correlate with ageing and hair damage: hairs from aged group tend to have damaged hairs in terms of topography and damaged hairs showed higher friction [19,29,30].

For topographical analysis, five body hairs from each mouse (n = 5) were cut and fixed to observation plates. Topographical data on the middle part of the fixed hairs were obtained by using a scanning probe microscope (SPM: SPM400, SII Nano-Technology, Japan). The damaged area on the hair surface which could correlate with aging [19,29] was evaluated from the topographical images using image processing software (ImageJ). Gaussian high-pass filtering with a 5 μm cut-off and cylindrical form removal were applied to facilitate visual inspection. The damaged area ratio was calculated as the damaged area divided by the total area.

With regard to the measurement of friction, a TL701 portable friction tester (Trinity-Lab Inc., Japan) was used. This tester allowed us to measure the friction on body hair directly without cutting or flattening the body hair of the mouse. A contact probe made of soft rubber was pressed onto the body hair of the forehead of each mouse. While applying a load of approximately 1 N, the probe was slid backward from the forehead across the area between the ears for 10 s, and the coefficient of friction (COF) was obtained. The COF was measured 10 times per mouse (n = 5), and the average value was evaluated.

### Statistical analysis

All values are expressed as means ± SEMs. Statistical analyses were performed using Tukey–Kramer tests for comparisons among multiple groups (Figs 2A–2C, 3C and 4A–4F), Dunnett tests for comparisons with the control groups (Figs 1C, 1D and 3B; S4, S5 and S6A Figs) and t-tests for comparisons between two groups (Fig 2D), using JMP 10.0.0 and GraphPad Software. Values of $^{*}P < 0.05$, $^{**}P < 0.01$, $^{***,\ \#\#\#}P < 0.001$, and $^{****,\ \#\#\#\#}P < 0.0001$ were considered to indicate statistical significance.

## Results

### Isolation of Naturido

We first prepared *I. japonica* grown on silkworm pupae (Fig 1A), *I. japonica* powder and *I. japonica* powder extract produced using the protocol described in the Methods section. We next used bioassay-guided screening to investigate astrocyte proliferation, because we first noticed useful effects of these substances on astrocyte proliferation as evidenced by BrdU incorporation (Fig 1D). Through two-phase distribution of the *I. japonica* extract, astrocyte proliferative assays and hydrophilic interaction liquid chromatography (HILIC)-HPLC (S1 Table), we obtained a final purified product that promoted an apparent astrocyte proliferation described below (Fig 1D), and analysed it with NMR and mass spectrometry (MS). The resulting cyclic peptide derivative was found to be a water-soluble cyclic peptide derivative with a molecular weight of 566.2588 (S1 Fig and S1 Results).

After searching for the structure of the cyclic peptide derivative in SciFinder, a database of chemical structures, the derivative was confirmed to be a novel compound (Fig 1B). We named the derivative Naturido (a combination of "Naturo" meaning nature and "id", a suffix of progeny in Esperanto).

### Direct effects of Naturido on astrocytes

To obtain essentially pure astrocyte cultures, we prepared primary cultures from mouse cerebral tissues [20]. Fluorescence imaging using antibodies for markers of neurons (MAP2), astrocytes (EAAT-2), microglia (CD11b/c), and oligodendrocytes (MBP) revealed that the

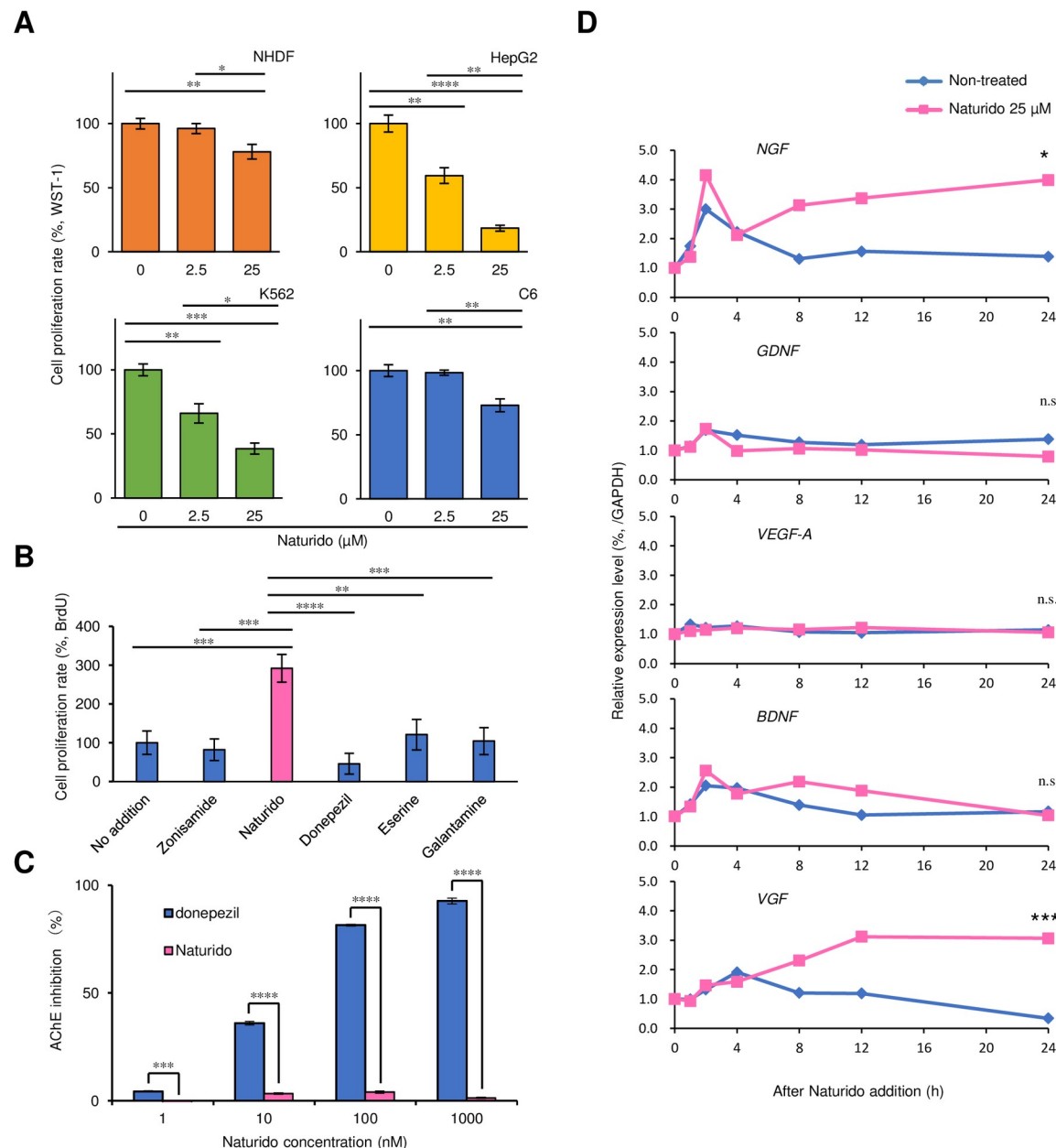

**Fig 2.** Relationship among Naturido and cell proliferation (A), therapeutic agents (B), acetylcholinesterase inhibitory activity (C), and the gene expression (D). (A) Normal human dermal fibroblasts (NHDFs), human hepatocellular carcinoma (HepG2) cells, human myeloid leukaemia (K562) cells and C6 glial tumour cells were used for proliferation assays with WST-1 (%) after treatment with Naturido at concentrations of 2.5 and 25 μM. (B) Each therapeutic agent (zonisamide, donepezil, eserine, and galantamine) was used at the same concentration as Naturido (25 μM), and proliferation was measured with a BrdU assay (%). (C) Comparison of the inhibitory effects of Naturido and donepezil on acetylcholinesterase (AChE) activity, shown as AChE inhibition (%). (D) The mRNA expression levels of representative neurotrophic factors (nerve growth factor = *NGF*, brain-derived neurotrophic factor = *BDNF*, vascular endothelial growth factor = *VEGF-A*, glial cell line-derived neurotrophic factor = *GDNF*, and non-acronymic neuropeptide = *VGF*) in astrocyte cultures were investigated to determine whether Naturido treatment enhances the expression of these genes. The Naturido concentration was 25 μM. All values are expressed as means ± SEMs. $^*P < 0.05$, $^{**}P < 0.01$, $^{***}P < 0.001$, $^{****}P < 0.0001$ vs each group (A-C Tukey-Kramer test using JMP 10.0.0) and vs the value at 24 h (D, t-test using JMP 10.0.0).

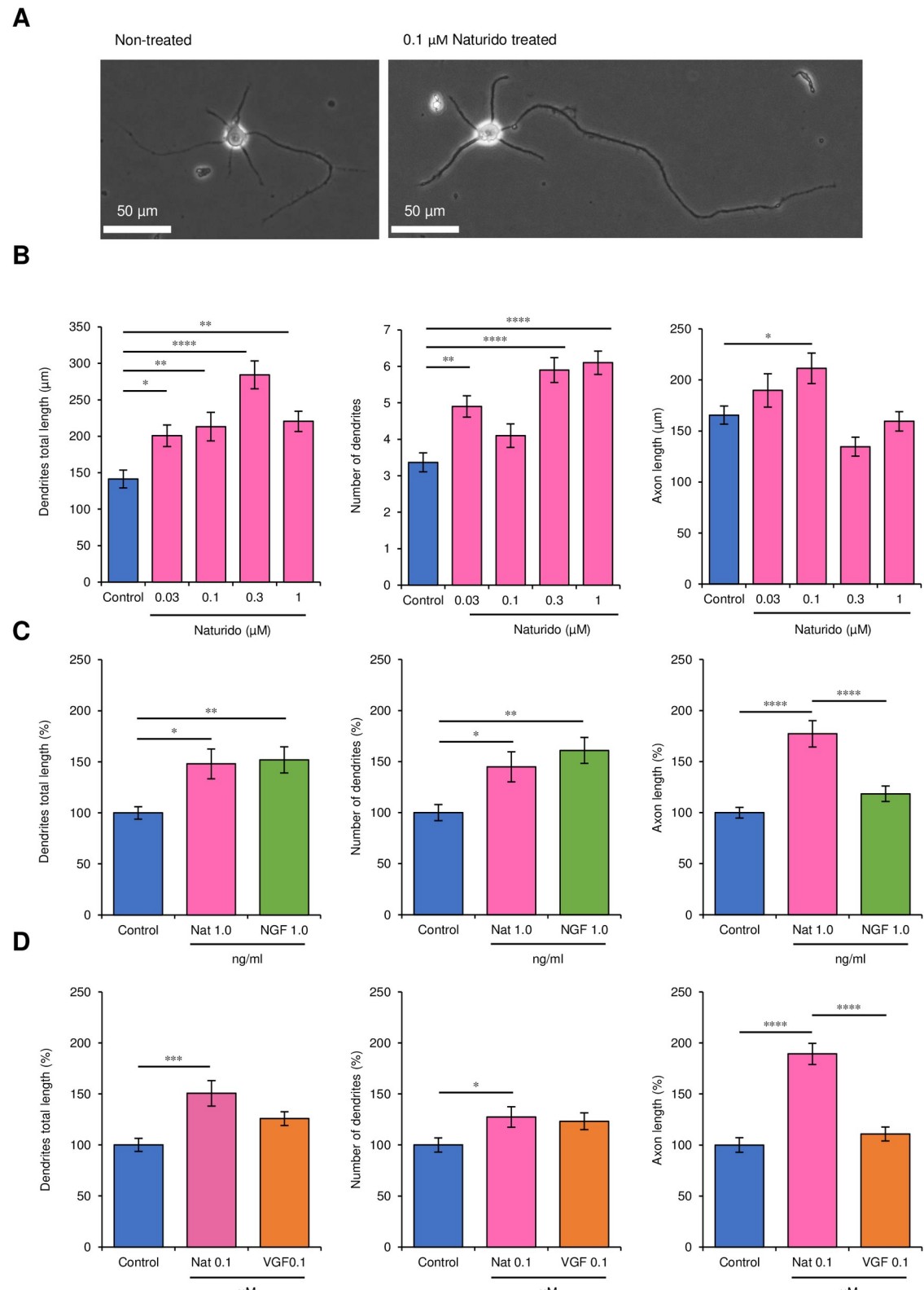

**Fig 3. Effects of Naturido on dendrite and axon development in hippocampal neurons.** (A) Representative photographs (left, non-treated; right, treated with 0.1 μM Naturido; scale = 50 μm); (B) Lengths of dendrites/neurons (μm, left), numbers of dendrites per neuron (middle), and lengths of axons/neurons (μm, right), as evaluated from 30 neurons after 3 days of the addition of Naturido at

concentrations from 0.03 μM to 1 μM. (C) Comparison of the effects of 1 ng/ml (= 0.0017 μM) Naturido and 1 ng/ml NGF protein on dendrite and axon development. Lengths of dendrites/neurons (μm, left), numbers of dendrites per neuron (middle), and lengths of axons/neurons (μm, right) and, as evaluated in 30 neurons. (D) Comparison of the effects of 0.1 μM Naturido and 0.1 μM VGF protein on dendrite and axon development. The observations and evaluations were conducted in the same way as in **c**. All values are expressed as means ± SEMs. *$P$ < 0.05, **$P$ < 0.01, ***$P$ < 0.001, **** $P$ < 0.0001 vs the control group (**b**, Dunnett test using JMP 10.0.0) or between groups (**c** and **d**, expressed as ratio of each control, Turkey-Kramer test using JMP 10.0.0).

astrocytes prepared in our cultures had a purity of 99% (EAAT-2 positive cells) or greater among the cerebral astrocytes.

As astrocyte dysfunction is associated with dementia and brain ageing but is further important area [12], we also compared the expression of GFAP (a marker that reflects astrocyte hypertrophy) and EAAT-2 (a marker that reflects altered glial function) in the studied astrocytes because double labelling shows that GFAP and EAAT2 do not entirely overlap [31]. As a result, about 25.8 to 39.8% of GFAP-positive cells within EATT-2 positive cells did not change significantly upon the addition of Naturido to the astrocyte culture medium (Fig 1C, vs before), indicating the stability of Naturido for the GFAP-positive populations with no the loss of EAAT-2 and no the increase of GFAP [31]. In the case of another astrocyte cultures with the addition of Naturido, no fluorescence imaging using antibodies for the markers of neurons (MAP2) and microglia (CD11b/c) was observed (S3 Fig). As GFAP is important for astrocyte–neuron interactions, and as astrocyte processes play vital roles in modulating synaptic efficacy in the CNS [32], the stable association between GFAP levels and Naturido suggests that this cyclic peptide may be useful in the analyses of astrocyte–neuron interactions. Indeed, the

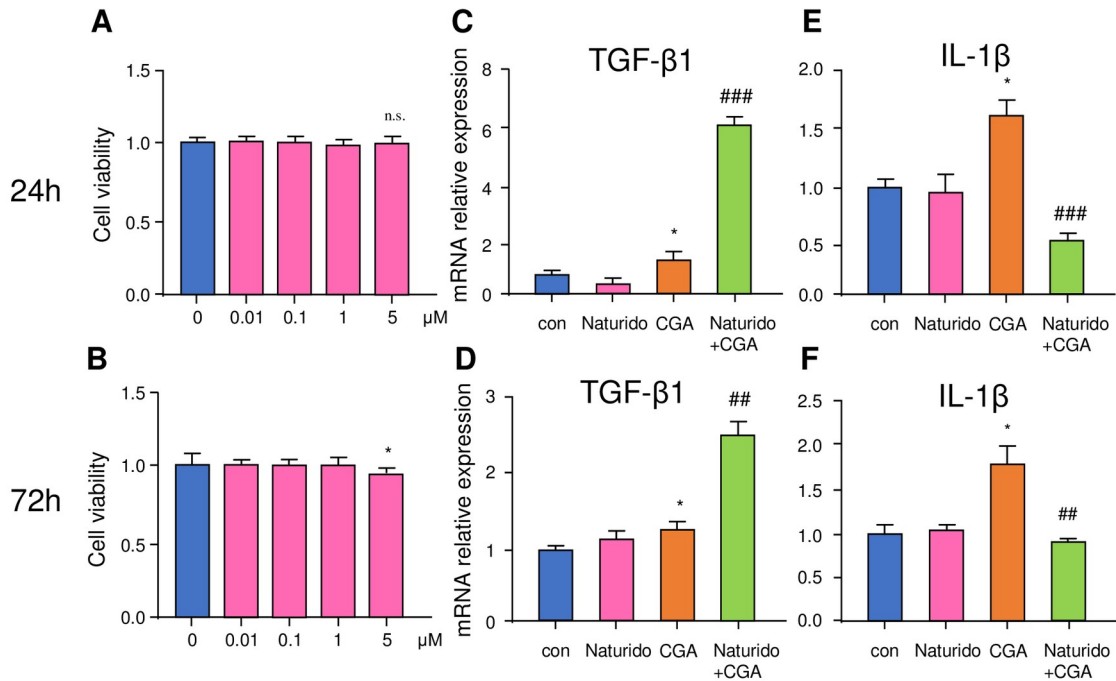

**Fig 4. Effects of Naturido on the expression of transforming growth factor (TGF)-β and interleukin (IL)-1β.** (A) and (B), Relationship between incubation time (a, 24 h; b, 72 h) and cell viability. (C) and (D), Effects of Naturido (1 μM), chromogranin A (CGA), or combined Naturido and CGA treatment for 24 h or 72 h on TGF-β expression. (E) and (F), Effects of Naturido (1 μM), chromogranin A (CGA), or combined Naturido and CGA treatment for 24 h or 72 h on IL-1β expression. All values are expressed as means ± SEMs. */#$P$ < 0.05, **/## $P$ < 0.01, ***/###$P$ < 0.001 vs the control (*) or CGA (#) group (Turkey-Kramer test using the GraphPad Prism software package).

measurement of BrdU incorporation showed that the addition of exogenous Naturido dose-dependently stimulated the proliferation of a clear proportion of astrocyte minority for GFAP and astrocyte majority for EAAT-2 (Fig 1D), although we did not evaluate other glial populations.

To investigate whether this Naturido-mediated promotion of proliferation was specific to astrocytes or was common to multiple cell types, we tested the effects of Naturido on four other cell lines about whether Naturido causes cancer: normal human dermal fibroblasts (NHDFs), human hepatocellular carcinoma (HepG2) cells, human myeloid leukaemia (K562) cells, and glial tumour (C6) cells. In contrast to astrocytes, the cell proliferation rates of all four cell types decreased significantly after the addition of Naturido (Fig 2A).

Given the astrocyte-specific characteristic of Naturido and its potential for use in the treatment of dementia, we compared the astrocyte proliferation induced by the addition of Naturido with that induced by treatment with zonisamide [17], donepezil [33], eserine [34], and galantamine [35], which are used in the clinical treatment of Parkinson's disease and AD. Zonisoamid [17], which increases C6 cells and is used in the clinical treatment of Parkinson's disease, had no effect on astrocyte proliferation. The other three drugs (donepezil, eserine, and galantamine) also did not have an effect on astrocyte proliferation. In contrast, Naturido modulated astrocyte proliferation. On the other hand, these three drugs have the acetylcholinesterase inhibitor function used for the clinical treatment of AD, but compared with donepezil, Naturido did not have this activity (Fig 2B and 2C). Given the results of this comparative assay (Fig 2), we hypothesized that Naturido might target astrocytes.

The expression of specific genes in astrocytes induced by the addition of Naturido suggests the possibility of an astrocyte-targeting function. Next, we investigated the effects of Naturido addition on the expression of five representative genes associated with synaptic strengthening and plasticity in astrocytes, as well as neuronal modulation [36–38]: *nerve growth factor* (*NGF*), *glial cell-derived neurotrophic factor* (*GDNF*), *vascular endothelial growth factor* (*VEGF-A*), *brain-derived neurotrophic factor* (*BDNF*), and *non-acronymic neuropeptide* (*VGF*). As shown in Fig 2D, the addition of Naturido to cultured astrocytes significantly stimulated a sustained increase in the mRNA levels of *NGF* and *VGF* relative to untreated control cells, but had no such effect on the levels of *GDNF*, *VEGF-A*, and *BDNF*. It is known that among medical drugs, that ifenprodil, an antagonist of NMDA receptor 2B, enhances the mRNA expression of the astrocytes and neurotrophic factor genes *NGF* and *BDNF* in cultured astrocytes [39], but the increases in the expression of *NGF* and *VGF* mRNA induced by Naturido addition in the same cultured cells suggest the induction of axon growth [40] and proliferation associated with hippocampal neurogenesis [41].

## Direct effects of Naturido on neurons

Next, to test whether Naturido targets neurons as well as glia, we investigated whether the peptide directly affected neuron growth. We chose hippocampal neurons, as they represent a major neurogenesis model, and used low concentrations of Naturido because neurons are more sensitive than astrocytes to Naturido (Fig 1D). The addition of low concentrations (0.03 to 1.0 μM) of Naturido to the culture medium of hippocampal neurons significantly increased the length and number of dendrites, except the number of dendrites at 0.1 μM; only axon length was notably increased by 0.1 μM Naturido, but axon length did not change at 0.3 and 1.0 μM Naturido (Fig 3A and 3B). We also compared the effects of treatment with Naturido with the effects of treatment with either NGF or VGF protein in cultured hippocampal neurons. In these assays, effects of Naturido on dendrite length and number were more or less inferior than NGF but its effects on axon length was more significant than NGF (Fig 3C). The

addition of Naturido also significantly increased dendrite length and number as well as axon length relative to VGF addition (Fig 3D). These findings suggest that Naturido represents a novel neurotrophic peptide in hippocampal neurons.

## Direct effects of Naturido on microglia

Building on our previous experiments focused on astrocytes and neurons, we next investigated whether Naturido influenced microglia functions. Microglia, which are the resident immune cells in the brain, play a critical role in controlling neuroinflammation in the context of neuro-degenerative diseases (including AD) and ageing by regulating pro- and anti-inflammatory cytokines [42]. Chromogranin A (CGA), a neurosecretory acidic glycoprotein neuron and which is accepted as a candidate activator of microglia for inducing cytokine production [43]. It is well known that interleukin (IL)-1β is a key pro-inflammatory cytokine [26] and transforming growth factor (TGF)-β1 is a prominent anti-inflammatory cytokine [44–46]. Accordingly, we investigated the effects of Naturido on the CGA-mediated upregulation of IL-1β and TGF-β1 in primary microglia. We found that TGF-β1 expression in cultured primary microglia was significantly enhanced both 24 h ($P < 0.05$) and 72 h ($P < 0.05$) after CGA treatment (Fig 4C and 4D). Surprisingly, compared with CGA treatment alone, pre-treatment with Naturido significantly increased TGF-β1 expression by 3-fold at 24 h ($P < 0.01$), and a 2-fold increase persisted at 72 h ($P < 0.01$) in the cultured microglia. In contrast, the CGA-induced increases in IL-1β expression were significantly attenuated at both 24 h ($P < 0/01$) and 72 h ($P < 0.01$) by pre-treatment with Naturido (Fig 4E and 4F). These observations indicate that Naturido shifts CGA-activated microglia towards an anti-inflammatory phenotype by increasing TGF-β1 and suppressing IL-1β expression. Notably, the fact that Naturido alone did not affect the expression of either TGF-β1 or IL-1β in primary microglia (Fig 4C and 4F) suggests that Naturido does not induce microglial activation in the brain. Therefore, pre-treatment with Naturido may be beneficial for preventing the excessive neuroinflammation induced by CGA by enhancing TGF-β1 and reducing IL-1β during AD and ageing.

## Effects of the oral administration of Naturido on learning and memory ability in senescence-accelerated mice

After assessing the effects of Naturido on astrocyte proliferation, neuron growth and anti-inflammation in microglia in primary cell cultures, and given our previous findings reported that the oral administration of the extract from *I. japonica* grown on silkworm pupae reversed astrogliosis in the hippocampus and improved memory deficits in aged mice [5,6], we next aimed to assess its effects in a mouse model of ageing. We investigated the effects of the oral administration of Naturido in mice with normal ageing (SAMR1 mice) and in senescence-accelerated model mice (SAMP8 mice), which have been used extensively [47]. Four groups were used to evaluate learning and memory: untreated control SAMR1 and SAMP8 mice, SAMP8 mice orally administered 1250 μg/kg/day donepezil (a typical treatment for AD [33]) and SAMP8 mice orally administered 2.5 or 25 μg/kg/day Naturido. Among the treatment groups, based on the latency observed during the post-shock trial in the passive avoidance test, SAMP8 mice administered donepezil showed poor contextual learning ability ($P < 0.01$) (S4 Fig). It was confirmed that in SAMP8 mice administered either concentration of Naturido, contextual learning ability recovered to the levels found in SAMR1 mice with normal ageing, but SAMP8 mice did not show a significant decrease in latency ($P = 0.06$). Because SAMP8 mice might be insensitive during the post-shock trial in the passive avoidance test, we carried out another assay, the Morris water maze test, to evaluate the spatial learning ability of SAMR1 and SAMP8 mice [5,6]. The percentage of the time spent within quadrant 0 (which had a

platform) was significantly lower for SAMP8 mice than for SAMR1 mice ($P < 0.05$) (S4 Fig). Additionally, a comparison of the SAMP8 mice with the SAMP8 mice orally administered Naturido (25 μg/kg/day) revealed that the percentage of time spent within quadrant 0 significantly recovered in the SAMP8 mice orally administrated Naturido, almost to the level observed for SAMR1 mice ($P < 0.05$). However, the oral administration of donepezil (1250 μg/kg/day) to SAMP8 mice did not result in recovery of the spatial learning ability to the same extent as the oral administration of Naturido (S5 Fig). Based on the results of both of these behavioural tests, we conclude that the oral administration of Naturido improves spatial learning ability in mice of age-related learning deficits.

### Effects of the oral administration of Naturido on hair quality in senescence-accelerated mice

As an animal model of accelerated senescence and ageing-associated disorders, SAMP8 mice have been used to evaluate the preventive effects of drugs [47]. Our previous paper showed that the deterioration of hair quality in SAMP8 mice which is a well-known characteristic of this mouse model, was improved by the oral administration of a nutraceutical (mulberry twig extract) [19]. In parallel with the effects of the oral administration of Naturido on learning and memory ability in SAMP8 mice, we investigated the relationship between the deterioration of hair quality in SAMP8 mice as a representative ageing-related marker [19] and the effects of Naturido administration. Hair quality was assessed by evaluating the correlation between the coefficient of friction (COF) and the damage area ratio for the body hair surface of the mouse, using the friction tester and SPM.

Dilapidated structures and some fine particle-like damages on the hair surface could be recognized visually especially for SAMP8 mice (S6B Fig). Such particle-like damages were also found for other samples. Dilapidated and particle-like structures were evaluated quantitatively based on the damaged area ratio defined as total damaged area divided by the evaluation area on the vertical axis of the S6A Fig.

The body hair of SAMP8 mice exhibited a high COF and damage area ratio value, while that of SAMR1 mice with normal ageing exhibited lower values for both indicators (S6A Fig). In SAMP8 mice orally administered donepezil, the damaged area of the observed hair surface was close to that of mice with normal ageing (SAMR1). However, the COF of the body hair of donepezil-treated SAMP8 mice was the same as that in untreated SAMP8 mice, and no significant reduction in the COF was observed. In contrast, in the case of SAMP8 mice orally administered either concentration of Naturido (2.5 or 25 μg/kg/day), both the COF and damage area ratio values were lowered to the same levels as those in mice with normal ageing; hair quality was improved following administration of Naturido (S6A amd S6B Fig). Thus, we believe that Naturido may exert an anti-ageing effect on hair.

### Discussion

In the brain science of this century, both sides of glia and neurons would be essential to proceed with new treatments of ageing-related deficits and neurodegenerative diseases [9]. Our previous evidence suggested that the powder or the extract of *I. japonica* might have a potential as hopeful candidate but the active component fulfilled with both sides of glia and neurons was not identified until today [5,6,8]. In this study, however, we found the novel cyclic peptide Naturido and elucidated that Naturido enhanced astrocyte proliferation, antiinflammation effects, and neuron growth *in vitro*. *In vivo* analyses also revealed that the oral administration of Naturido reversed ageing-related deficits in senescence-accelerated mice. It remains how Naturido acts on CNS by the oral administration, but it is inferred that Naturido brings about

bidirectional interactions between the gut and the brain axis [48]. In summary, the results of these *in vitro* and *in vivo* experiments suggest that the novel cyclic peptide Naturido is a valuable and promising therapeutic agent for use in the development of innovative treatments affecting glia-neuron interactions associated with CNS disorders.

Published studies with important insights into both glia and neurons include reports on tripartite synapses that comprise both glia and neurons [10] and astrocyte-synapse interactions in CNS disorders [11]. New and extensive functions for glia have also been elucidated: the stem cell potential of astrocytes [7], the role of astrocytes in AD and age-associated dementias [12], the role of astroglia in AD [49], the multi-faceted activities of microglia [13], the role of microglial disease [50] and the presence of microglia throughout the lifespan [51]. However, no drugs have yet been developed from these important insights.

Many physiological compounds used in the treatment of diseases have been identified from fungi. For example, a novel synthetic compound (FTY720) with low toxicity and *in vivo* immunosuppressive activity was developed from a fungus (*I. sinclairii*) metabolite, the primary compound of which was myriocin [52]. This synthetic compound, fingolimod, has provided a new approach for the treatment of multiple sclerosis [53]. Fingolimod that originates from a fungus may be model for drug development. In our pilot study [8], the fungus powder was used as a nutraceutical for AD patients, and the oral administration of this nutraceutical significantly increased the acetylcholine concentration in cerebrospinal fluid. Thus, for practicalities of clinical development, we could try a mass production of the cultivation of the fungus and might be feasible to produce Naturido by chemical synthesis.

The three main findings of the *in vitro* cell experiments and the *in vivo* mouse experiments in the present study and in our previous human study, astrocyte proliferation, shifts in CGA-activated microglia phenotypes, and neuron growth, suggest that the involvement of Naturido in species-common cell receptors and glia–neuron interactions may have novel applications. It remains to be determined whether this cyclic peptide crosses the blood–brain barrier [54] and modulates oligodendrocytes [55], which are the third type of glial cell, and the specific receptors in astrocytes, microglia, and neurons have yet to be identified. However, we believe that Naturido will provide a tool for the clarification of the multidirectional chemical crosstalk involved in glia-neuron interactions. Recently, in the case of the pilocarpine or kainic acid-induced status epilepticus rat model, effects of metalloproteinase-12 inhibition on astrocytes, microglia and neurons in the CA3 hippocampal region are assessed and the specific metalloproteinase-12 inhibition means a potential therapeutic agent against neurological disorders [56,57]. Naturido and the combination with Naturido and metalloproteinase inhibition also may be responsible for the treatment of status epilepsy. In conclusion, the results of the present study and our previous pilot study on humans will undoubtedly support the development of therapeutic strategies for the treatment of neural diseases. We hope that Naturido itself will represent a candidate for a glia/neuron-targeting drugs, warranting further studies on the molecular-chemical crosstalk between glia and neurons and its use in the clinic.

## Supporting information

**S1 Fig. Experimental schedule of mice prepared for *in vitro* and *in vivo* experiments.** (TIF)

**S2 Fig. Chemical formula of the F3-10-4-5-3 fraction.** (A) After purification, the structure of F3-10-4-5-3 demonstrating astrocyte proliferation was considered a bis (diethylamine) salt ([α] D −21.6 (*c* 0.18, H2O)). The molecular formula for @ was established as C26H38N4O10. The parameters were as follows: highresolution electrospray ionization mass spectroscopy [HRESIMS]: fast atom bombardment (FAB) negative; matrix: glycerol; high-resolution mass

spectrometry (HRMS) (FAB) *m/z* (M-H)-: calculated for [C26H37N4O10-H]-, 565.2510, found, 565.2512. The 1H NMR spectrum (in D2O, (B) showed resonances for six methine protons (δH 4.80, 4.75, 4.12, 4.03, 3.88, and 1.84 ppm), three methyl protons (δH 2.72, 1.67, and 1.03 ppm), characteristic upfield methyl groups (δH 0.71 and 0.81 ppm), and three aromatic protons of a 1,2,4-trisubstituted benzene (δH 7.15, 7.04, and 6.97 ppm). The 13C NMR spectrum showed five carbonyl carbons (δC 184.9, 181.0, 176.2, 173.9, and 173.0 ppm), six aromatic carbons (δC 153.5, 145.1, 134.6, 125.2, 124.2, and 121.4 ppm), and 15 carbons (δC 88.2, 77.6, 74.5, 62.4, 62.1, 58.4, 37.1, 36.4, 35.1, 31.8, 31.0, 23.3, 21.1, 20.7, and 10.4 ppm) in the aliphatic region. Further analysis of 1D and 2D NMR data indicated the 2 presence of four substructures (C). The structure of I was established to be β-hydroxy-3,4-dihydroxyphenylalanine by heteronuclear multiple bond correlation (HMBC) analysis, which indicated a correlation of H-10 with C-12, C-16 and C-8 and of 9-NMe with C-9. Partial structure II was revealed as a hydroxyl isoleucine moiety on the basis of the HMBC correlations of both H-21 and H-22 with C-3 and of H-23 with C-2 as well as the 13C chemical shift at C-2 (δC 85.6 ppm). Partial structures III and IV were rapidly classified as valine and glutamic acid, respectively, by 1H-1H correlation spectroscopy (COSY). (D) Cyclic peptide consisting of 4 amino acids (N-methyl-β-hydroxy DOPA, valine, β-hydroxyleucine, and glutamic acid).
(TIF)

**S3 Fig. Histochemical observations of neurons and microglia in astrocyte cultures treated with Naturido.** According to the sections of cell culture experiments and identification of primary cultured astrocytes in "materials and methods", the astrocytes with a density of $5.6 \times 10^3$ cells/cm$^2$ were prepared and the cells were exposed to Naturido (25 μM) for 0 or 24 h under of 37˚C and 5.0% CO$_2$. LG-D-MEM (0% FBS) without the addition of Naturido was used as a control. Histochemical observations were carried out using DAPI for nuclear staining, anti-MAP2 for a neuronal marker, anti-CD11b/c for a microglia marker, and anti-EAAT-2 for an astrocyte marker. (A) Histochemical observations (left panel) of anti-MAP2 in the primary cultured astrocytes (before, 0 h, 24 h treated with Naturido) compared to the stains of trichrome, DAPI and anti-EAAT-2. (B) Histochemical observations (right panel) of anti-CD11b/c in the primary cultured astrocytes (before, 0 h, 24 h treated with Nautrido) compared to the stains of trichrome, DAPI and anti-EAAT-2. Scale bar = 100 μm.
(TIF)

**S4 Fig. Effects of Naturido on the step-through passive avoidance learning test in senescence-accelerated mouse (SAMP8) mice.** The mice were divided into five treatment groups, mice with normal ageing (SAMR1 mice) (n = 7) served as vehicle controls with oral administration of saline (0.9%) for 5 weeks, SAMP8 mice (n = 14) with oral administration of saline (0.9%) for 5 weeks, SAMP8 + donepezil HCl (1250 μg/kg/day) mice (n = 14) administered donepezil HCl (1250 μg/kg/day) orally for 5 weeks, SAMP8 + Naturido (2.5 μg/kg/day) mice (n = 16) administered Naturido orally at a dosage of 2.5 μg/kg/day for 5 weeks, SAMP8 +Naturido (25 μg/kg/day) mice (n = 15) administered Naturido orally at a dosage of 25 μg/kg/day for 5 weeks. All values are expressed as means ± SEMs. $^{**}P < 0.01$ vs the SAMP8 group (Dunnett test using JMP 10.0.0).
(TIF)

**S5 Fig. Effects of Naturido on the time spent in the target quadrant and in other quadrants in a Morris water maze test with senescence-accelerated model mice (SAMP8 mice).** (A) The times spent in the target quadrant and in the other quadrants were compared on day 9, mice with normal ageing (SAMR1 mice) (n = 7), Senescence-accelerated mouse (SAMP8) mice (n = 12), SAMP8 + donepezil HCl (1250 μg/kg/day) mice (n = 11), SAMP8 + Naturido (2.5 μg/kg/day) mice (n = 14), SAMP8 + Naturido (25 μg/kg/day) mice (n = 7). (B)

Representative images of the circular swimming path captured by video. Left, senescence-accelerated mice (SAMP8 mice); right, SAMP8 + Naturido (25 μg/kg/day) mice. All values are expressed as means ± SEMs. $^*P <$0.05 vs the SAMP8 group (Dunnett test using JMP10.0.0).
(TIF)

**S6 Fig. Effects of Naturido on the coefficient of friction (COF) and body hair damage in senescence-accelerated mice.** (A) Measurements (n = 5 in 5 experimental groups) were taken using a static and dynamic friction tester and a scanning probe microscope (SPM), and the COF (n = 5 in 5 experimental groups) was plotted against the damaged area divided by the total area. All values are expressed as the means ± SEMs. $^{****}P <$0.0001 and $^{####}P <$0.0001 vs the SM group ($^*$SPM, $^#$COF) (Dunnett test using JMP 10.0.0). (B) Representative SPM images of damaged areas in each mouse group. §A severely damaged area in an SAMP8 mouse. The upper right column shows the height of the coloured bar.
(TIF)

**S1 Table. Purification process of the components promoting astrocyte proliferation from the hot water extract of a fungus (*Isaria japonica*) grown on the silkworm *Bombyx mori.***
(A) Beginning with 42 g of fungus powder, a purification procedure was carried out according to the flow chart, and physiological activity was tested with an astrocyte proliferation assay. F3-10-4-5-3 was confirmed as the final fraction promoting astrocyte proliferation, and the recovery of this final fraction was estimated to be 0.03% (12.6 mg) (B). Elution results from high-performance liquid chromatography (HPLC)/hydrophilic interaction chromatography (HILIC) with columns are shown in S1A Fig.
(TIF)

**S2 Table. Sequences of the primer pairs.** Analysis of gene expression was carried out by using the synthesized cDNA as a template, commercially available primers (obtained from TaKaRa), SYBR premix Ex Taq I (TaKaRa) and a real-time PCR device, the Thermal Cycler Dice® TP800 TaKaRa). The primers comprised the following: a primer for *Mus musculus Ngf* transcript variant 1 mRNA (MA07578), a primer for *Mus musculus Gdnf* mRNA (MA102345), a primer for *Mus musculus Vegfa* transcript variant 1 mRNA (MA128545), a psrimer for *Mus musculus Bdnf* transcript variant 2 mRNA (MA138332), and a primer for *Mus musculus Vgf* mRNA (MA157656). The expression level of each target gene was compared after calibration against an internal standard, the housekeeping gene *GAPDH*.
(TIF)

**S1 Results. Chemical formula of the F3-10-4-5-3 fraction.**
(TIF)

**S1 Data.**
(XLSX)

## Acknowledgments

We are grateful to D. Yoshida, the honorary chairman; T. Shudo, the president; K. Kumagai, the general affairs manager; M. Kumagai, the illustrator; and C. Ebata, the research assistant, for their thoughtful encouragement and financial support. We also thank R. Sato of Esperanto, who named Naturido.

## Author Contributions

**Conceptualization:** Koichi Suzuki.

**Data curation:** Shinichi Ishiguro, Tetsuro Shinada, Zhou Wu.

**Formal analysis:** Shinichi Ishiguro, Tetsuro Shinada, Zhou Wu.

**Funding acquisition:** Koichi Suzuki.

**Investigation:** Shinichi Ishiguro, Mayumi Karimazawa, Michimasa Uchidate, Eiji Nishimura, Yoko Yasuno, Makiko Ebata, Piyamas Sillapakong, Hiromi Ishiguro, Nobuyoshi Ebata, Junjun Ni, Muzhou Jiang, Masanobu Goryo.

**Methodology:** Shinichi Ishiguro, Tetsuro Shinada, Zhou Wu, Michimasa Uchidate, Masanobu Goryo, Keishi Otsu, Hidemitsu Harada.

**Project administration:** Koichi Suzuki.

**Resources:** Shinichi Ishiguro, Tetsuro Shinada, Zhou Wu, Mayumi Karimazawa, Michimasa Uchidate, Hiromi Ishiguro, Hidemitsu Harada.

**Supervision:** Tetsuro Shinada, Zhou Wu, Koichi Suzuki.

**Validation:** Shinichi Ishiguro, Tetsuro Shinada, Zhou Wu, Koichi Suzuki.

**Visualization:** Shinichi Ishiguro, Tetsuro Shinada, Zhou Wu, Koichi Suzuki.

**Writing – original draft:** Shinichi Ishiguro, Tetsuro Shinada, Zhou Wu, Mayumi Karimazawa, Koichi Suzuki.

**Writing – review & editing:** Shinichi Ishiguro, Tetsuro Shinada, Zhou Wu, Mayumi Karimazawa, Koichi Suzuki.

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
