## [Decision Letter · Decision Letter 0]

30 Sep 2020

PONE-D-20-19724

A novel cyclic peptide (Naturido) modulates glia–neuron interactions in vitro and reverses ageing-related deficits in senescence-accelerated mice

PLOS ONE

Dear Dr. Suzuki,

Thank you for submitting your manuscript to PLOS ONE. After careful consideration, we feel that it has merit but does not fully meet PLOS ONE’s publication criteria as it currently stands. Therefore, we invite you to submit a revised version of the manuscript that addresses the points raised during the review process.

We look forward to receiving your revised manuscript.

Kind regards,

Giuseppe Biagini, MD

Academic Editor

PLOS ONE

Journal Requirements:

2.Thank you for stating the following in the Financial Disclosure section:

[YES

KS, JSPS KAKENHI Grant No. 23228001].   

We note that one or more of the authors are employed by a commercial company: Biococoon Laboratories, Inc

Reviewers' comments:

Reviewer's Responses to Questions

**Comments to the Author**

1. Is the manuscript technically sound, and do the data support the conclusions?

Reviewer #1: Yes

Reviewer #2: Yes

2. Has the statistical analysis been performed appropriately and rigorously? 

Reviewer #1: Yes

Reviewer #2: Yes

3. Have the authors made all data underlying the findings in their manuscript fully available?

Reviewer #1: Yes

Reviewer #2: Yes

4. Is the manuscript presented in an intelligible fashion and written in standard English?

Reviewer #1: Yes

Reviewer #2: Yes

5. Review Comments to the Author

Reviewer #1: The manuscript brings original and robust characterization of a peptide Naturido in neurons and glia response and viability in vitro and in the behavior in vivo using the senescence-accelerated mouse model.

However, some minor and major points need to be revised, as follows.

Methos:

Line 248 Please check some typing errors. i.e. include a space between words (24h)

Line 309 - The final cellular density (concetration) in purified astrocyte cultures for 96 wells plates are expressed as f 2.0 × 10�5 cells/ml. Please include information of cells/cm2

Line 318 please include Naturido concentrations adopted in this study

Include the choice of Naturido concentration (25 µM) adopted in subsequent experiments

Please include minimum of information about the inhibitory activity on acetylcholinesterase assay adopted as cell number and density (cells/cm2) and time experiments

Line 402 For neuronal cultures include cell number and density (cells/cm2). It is no clear “ After 3 h on the cell seeding, each coverslip was gently inverted with fine forceps and 403 transferred to a 12-well plate, the wells of which already contained 1 ml each of 404 Neurobasal/B27 and the test agent (Naturido)”

Include replicates performed for all in vitro experiments.

Line 409 In the section “Proliferation test for cells other than astrocytes and neurons” include cell density (cells/cm2) for all cell lines adopted and plates adopted.

Line 422 In the section “Analysis of astrocyte gene expression” include astrocyte density (cells/cm2) and plates adopted.

Line 462 In the section “Determination of cell viability”, please correct to “Determination of microglia viability”. Include microglia density (cells/cm2) adopted.

Line 473 RT-PCR analysis, please correct to “Microglia RT-PCR analysis”. Similar experiments were adopted to microglia and astrocytes and information needs to be clear. Include microglia density (cells/cm2) adopted.

Line 492 to In in vivo experiments, please justify 2.5 µg and 2.5 µg Naturido/kg/day concentrations adopted

Results

Figure 1 C. Apparently and surprising you found a great proportion astrocyte negative for GFAP (shown in green) and the majority positive for EAAT1 (shown in red). How could you explain it. Cells. Moreover, immunocytochemistry and negative antibody controls for microglia and neurons must be included.

Discussion

Naturido was tested in cells of different species (mouse, rat and humans). As it is a peptide what about cellular target and specie specificity?

Moreover, considering that Naturido treatment in mouse was orally, how could you explain effects on central nervous system? I think the results could be better discussed.

Reviewer #2: Authors obtained quite interesting findings; however, the following points need to be deeply improved:

• “However, the because glia–neuron interactions are extremely complex and because individual astrocytes or microglia may be targeted, the development of effective preventive and therapeutic agents for AD and other neurological disorders represents a significant challenge.” Please rephrase this sentence.

• Please specify the total number of animals used and the number of animals used for each experiment.”

• In legend to figure, N° animals/group adopted for each technique should be provided.

• Why, in your opinion, is the GFAP positive cell ratio not significant in comparison to the non-treated group and control group (Figure 1C)?

• Why is the SEM bar absent at 0 um (Figure 1D)?

• Do you think Naturido could be used also after pilocarpine (Vinet et al., 2018) or kainic acid- (Costa et al., 2020) induced status epilepticus, in which neurons and astrocytes are usually affected? Could Naturido be used in combination to the hydroxypyrone-based inhibitor of metalloproteinase-12 to enhance its effect?

References:

1. Vinet et al. (2018) A hydroxypyrone-based inhibitor of metalloproteinase-12 displays neuroprotective properties in both status epilepticus and optic nerve crush animal models. International Journal of Molecular Sciences. doi: 10.3390/ijms19082178

2. Costa et al. (2020) Status epilepticus dynamics predicts latency to spontaneous seizures in the kainic acid model. Cell Physiol Biochem. doi: 10.33594/000000232

6. PLOS authors have the option to publish the peer review history of their article (what does this mean?). If published, this will include your full peer review and any attached files.

Reviewer #1: No

Reviewer #2: No

---

## [Author Response · Author response to Decision Letter 0]

23 Nov 2020

We have exerted all possible efforts for excellent suggestons by reviewers# 1 and 2. We would like to expect a good information from you.

---

## [Decision Letter · Decision Letter 1]

28 Dec 2020

A novel cyclic peptide (Naturido) modulates glia–neuron interactions in vitro and reverses ageing-related deficits in senescence-accelerated mice

PONE-D-20-19724R1

Dear Dr. Suzuki,

We’re pleased to inform you that your manuscript has been judged scientifically suitable for publication and will be formally accepted for publication once it meets all outstanding technical requirements.

Kind regards,

Giuseppe Biagini, MD

Academic Editor

PLOS ONE

Additional Editor Comments (optional):

Reviewers' comments:

Reviewer's Responses to Questions

**Comments to the Author**

1. If the authors have adequately addressed your comments raised in a previous round of review and you feel that this manuscript is now acceptable for publication, you may indicate that here to bypass the “Comments to the Author” section, enter your conflict of interest statement in the “Confidential to Editor” section, and submit your "Accept" recommendation.

Reviewer #2: All comments have been addressed

2. Is the manuscript technically sound, and do the data support the conclusions?

Reviewer #2: (No Response)

3. Has the statistical analysis been performed appropriately and rigorously? 

Reviewer #2: (No Response)

4. Have the authors made all data underlying the findings in their manuscript fully available?

Reviewer #2: (No Response)

5. Is the manuscript presented in an intelligible fashion and written in standard English?

Reviewer #2: (No Response)

6. Review Comments to the Author

Reviewer #2: All comments have been addressed. I have no further comments. The article can be consider for publication.

7. PLOS authors have the option to publish the peer review history of their article (what does this mean?). If published, this will include your full peer review and any attached files.

Reviewer #2: No

---

## [Editor Report · Acceptance letter]

4 Jan 2021

PONE-D-20-19724R1 

A novel cyclic peptide (Naturido) modulates glia–neuron interactions *in vitro* and reverses ageing-related deficits in senescence-accelerated mice 

Dear Dr. Suzuki:

I'm pleased to inform you that your manuscript has been deemed suitable for publication in PLOS ONE. Congratulations! Your manuscript is now with our production department. 

Kind regards, 

on behalf of

Dr. Giuseppe Biagini 

Academic Editor

PLOS ONE